# The Diversity, Resistance Profiles and Plasmid Content of *Klebsiella* spp. Recovered from Dairy Farms Located around Three Cities in Pakistan

**DOI:** 10.3390/antibiotics12030539

**Published:** 2023-03-08

**Authors:** Samia Habib, Marjorie J. Gibbon, Natacha Couto, Khadija Kakar, Safia Habib, Abdul Samad, Asim Munir, Fariha Fatima, Mashkoor Mohsin, Edward J. Feil

**Affiliations:** 1The Milner Centre for Evolution, Department of Life Sciences, University of Bath, Bath BA2 7AY, UK; 2Centre for Genomic Pathogen Surveillance, Big Data Institute, University of Oxford, Oxford OX3 7LF, UK; 3Department of Biotechnology, Faculty of Life Sciences & Informatics, Balochistan University of Information Technology, Engineering and Management Sciences, Quetta 08763, Pakistan; 4Sardar Bahadur Khan Womens’ University, Quetta 08763, Pakistan; 5Center for Advanced Studies in Vaccinology & Biotechnology (CASVAB), University of Balochistan, Quetta 08763, Pakistan; 6Institute of Microbiology, University of Agriculture, Faisalabad 38000, Pakistan

**Keywords:** *Klebsiella pneumoniae*, One-Health, AMR, plasmids, Pakistan, agriculture

## Abstract

The rise of antimicrobial resistance (AMR) in bacterial pathogens such as *Klebsiella pneumoniae* (Kp) is a pressing public health and economic concern. The ‘One-Health’ framework recognizes that effective management of AMR requires surveillance in agricultural as well as clinical settings, particularly in low-resource regions such as Pakistan. Here, we use whole-genome sequencing to characterise 49 isolates of *Klebisella* spp. (including 43 Kp) and 2 presumptive *Providencia rettgeri* isolates recovered from dairy farms located near 3 cities in Pakistan—Quetta (*n* = 29), Faisalabad (*n* = 19), and Sargodha (*n* = 3). The 43 Kp isolates corresponded to 38 sequence types (STs), and 35 of these STs were only observed once. This high diversity indicates frequent admixture and limited clonal spread on local scales. Of the 49 *Klebsiella* spp. isolates, 41 (84%) did not contain any clinically relevant antimicrobial resistance genes (ARGs), and we did not detect any ARGs predicted to encode resistance to carbapenems or colistin. However, four Kp lineages contained multiple ARGs: ST11 (*n* = 2), ST1391-1LV (*n* = 1), ST995 (*n* = 1) and ST985 (*n* = 1). STs 11, 1391-1LV and 995 shared a core set of five ARGs, including *bla*_CTX-M-15_, harboured on different AMR plasmids. ST985 carried a different set of 16 resistance genes, including *bla*_CTX-M-55_. The two presumptive *P. rettgeri* isolates also contained multiple ARGs. Finally, the four most common plasmids which did not harbour ARGs in our dataset were non-randomly distributed between regions, suggesting that local expansion of the plasmids occurs independently of the host bacterial lineage. Evidence regarding how dairy farms contribute to the emergence and spread of AMR in Pakistan is valuable for public authorities and organizations responsible for health, agriculture and the environment, as well as for industrial development.

## 1. Introduction

The rise in antimicrobial resistance (AMR) is of pressing concern for public health and food security, and the role of non-clinical settings in the emergence of AMR (‘One-Health’) has been under increasing scrutiny [1,2]. This perspective is most pertinent in low-resource settings where there is a relatively high level of contact between humans and animals, and where antibiotic usage in agriculture is poorly regulated [3]. A recent report from the UN Environment Program [4] summarized the current evidence concerning the impact of antimicrobial resistance on human health and, specifically, the role of environmental drivers in the development and transmission of AMR between humans and animals. These drivers include antimicrobial usage, microbial diversity and anthropologic factors, in particular, the extent of sanitation infrastructure [5]. The UN report [4] also identifies three economic sectors (pharmaceuticals and other chemicals, agriculture and food, and healthcare) where effective monitoring, disclosure and transparency are critical for targeted interventions and for realigning incentives for AMR management.

Pakistan is an important case in point; more than 70% of the population of Balochistan, in the west of the country, are directly or indirectly involved in raising animals, and large-scale livestock farming is common, contributing to over 60% of agricultural output and over 11% of GDP nationally [6,7]. Moreover, farmers in Pakistan routinely use antibiotics as dietary supplements for livestock [8,9,10], which can accelerate the emergence and spread of multidrug-resistant (MDR) bacteria in agricultural settings. In addition to commercial losses, this has public health implications as the MDR strains may infect humans, for example, through contaminated food products, and potentially go on to cause outbreaks in healthcare settings. Insufficient monitoring of antibiotic usage and poor molecular surveillance of antimicrobial resistance in Pakistan has led to a paucity of data on the scale of AMR in a ‘One Health’ context and a weak evidence-base for targeted intervention measures [8].

*Klebsiella pneumoniae* (Kp) commonly colonises the guts of humans and animals as an asymptomatic commensal [11,12]. However, Kp is also an opportunistic pathogen that can cause serious infections in humans and commercially important infection in cows and other livestock. For example, Kp is a common cause of bovine mastitis, which results in significant commercial losses due to the deterioration of the taste, colour and odour of milk. Kp has also been isolated from other animal hosts, including companion animals, poultry, invertebrates and wild birds [13,14].

A key challenge in the fight against AMR is the management of resistance against third generation cephalosporins and carbapenems, conferred by genes encoding extended-spectrum β-lactamases (ESBL) or carbapenemases, respectively. Carbapenem-resistant Kp (CRKP) isolates harbouring *bla*_NDM-1_ and *bla*_OXA-48_ genes from clinical, environmental and animal sources have previously been detected in Pakistan [15]. However, whole-genome sequencing (WGS) has not previously been applied to characterise Kp isolates from dairy cattle in Pakistan, and there is almost no evidence regarding the diversity, spread and plasmid content of the Kp population in this context.

In this study, we aimed to generate data on the prevalence and distribution of clinically relevant antimicrobial resistance genes (ARGs) within isolates of *Klebsiella* spp. isolated from healthy dairy cattle representing distinct geographical regions in Pakistan. We generated WGS data for 49 *Klebsiella* genomes (43 Kp) and 2 isolates of *Providencia rettgeri* from dairy cattle. The samples were taken from 10 farms located around 3 cities in Pakistan: Faisalabad, Sargodha and Quetta.

## 2. Methods

### 2.1. Sampling

A total of 150 cattle rectal samples were collected in sterile charcoal swabs from 10 dairy farms with an average herd size of *n* > 100 in Faisalabad, Sargodha and Quetta in Pakistan over a one-month period from January–February 2022. Samples were collected randomly from healthy, lactating dairy cattle. The samples were shipped under refrigerated conditions to the AMR Research lab, Institute of Microbiology, University of Agriculture Faisalabad for culturing and initial processing.

### 2.2. Isolation of Klebsiella spp. and Whole-Genome Sequencing

Samples were enriched in BHI broth supplemented with amoxicillin at concentration of 10 µg/mL for overnight at 37 °C. The enriched samples were streaked on Simmons citrate agar supplemented with amoxicillin and myo-inositol at concentration of 10µg/mL and 10%, respectively (Sigma; SCAI; [16]). Amoxycillin was used to select for *Klebsiella* spp. following Thorpe et al. [14]; we did not select for other resistance phenotypes. The plates were incubated at 37 °C for 48 h. After incubation, presumptive *Klebsiella* colonies were identified on the basis of having a bright yellow colour. One colony per sample was further selected and confirmed via cultivation on UTI ChromoSelect agar (sigmaaldrich.com) at 37 °C for 24 h. After incubation, blue to purple-coloured colonies indicated the growth of *Klebsiella*. A small number of colonies were randomly chosen and further confirmed using the Vitek-2 identification system (https://www.biomerieux-usa.com/ accessed on 1 February 2022). Pure *Klebsiella* colonies were transferred to the UK with a charcoal swab for further analysis. Only a single colony from each sample was selected for sequencing.

Sequencing was carried out on the Illumina platform by MicrobesNG (Birmingham, UK), and run through their standard analysis pipeline (including genome assembly using SPAdes 3.14.4. For full protocols, including DNA extraction, see MicrobesNG Whole Genome Sequencing Service Methods. The quality of the assemblies was verified using different parameters: GC content, number of contigs, total length of assembly and N50.

### 2.3. Genome Characterization

Genome assemblies that passed QC were analysed using Kleborate v0.4.0-beta [17] to assign species and multilocus sequence type, and screened for virulence and resistance genes. Abricate v0.9.8 (https://github.com/tseemann/abricate accessed on 1 February 2022) was used for further screening for resistance genes in the ResFinder database [18] (downloaded 17 August 2022) and virulence factors in the vfdb database (downloaded 18 August 2022). We scored the presence or absence of AMR and virulence genes using a threshold of >80% nucleotide identity and coverage. Short reads were mapped to the genome of Kp isolate PAK-014, which we previously isolated from hospital wastewater in Quetta (unpublished) using Snippy v4.3.6 (https://github.com/tseemann/snippy accessed on 1 February 2022). Snippy results were used as input for FastTree v2.1.11 [19,20] to generate an approximate maximum-likelihood phylogenetic tree. The phylogenetic tree and associated metadata were visualised using Microreact v23.0.0 [21]. Kleborate output was visualised using Kleborate-viz [17].

To determine the plasmid content of the isolates, we used MOB-suite [22] to classify contigs as plasmid-borne or chromosomal. This program uses Mash distances to assign contigs to plasmids according to a closed reference database. We used the default parameters that are already optimised for *Enterobacteriaceae* plasmids. This approach identifies the accession number of the plasmid with the shortest Mash distance to a given set of contigs but, depending on the database, we recognize that substantial size or structural variation may still be present between the query contigs and the returned plasmid.

## 3. Results

### 3.1. Species Assignment

Genomes of 55 presumptive Kp from dairy cows were sequenced as described in Methods. Of these, 4 isolates were excluded because of low-quality assembly and 51 assemblies were taken forward for further analysis (summarised in Table 1 and Appendix A). More detail, including the full Kleborate output, the tree and geographical information (including a map) is available via the Microreact project at https://tinyurl.com/37xn4m62, accessed on 6 January 2023. The 51 sequenced isolates were obtained from 7 separate dairy farms in Quetta (*n* = 29), in the west of Pakistan, close to the border with Afghanistan, from 2 dairy farms in Faisalabad in the northeast (*n* = 19), and from one in Sargodha (*n* = 3), which lies around 100 Km to the northwest of Faisalabad. In total, 43 of the genome assemblies were assigned by Kleborate as Kp: 24/29 from Quetta, 16/19 from Faisalabad and 3/3 from Sargodha.

Of the eight isolates that were not identified as Kp, four were *K. similipneumoniae* (three from Quetta, one from Faisalabad) and two were *K. variicola* (both from the same dairy farm in Faisalabad). We also sequenced two isolates, both from the same farm in Quetta, that were assigned as *P. rettgeri* and *Citrobacter amalonaticus*. Whilst the Kleborate species assignment for the *P. rettgeri* isolate is ‘strong’, and the assembly is of high quality (N50 = 792,114, contig count = 66), for the *C. amalonticus* isolate the species assignment is ‘Weak’, and the quality of the assembly is lower (N50 = 63,914, contig count = 469). Moreover, the total assembly size for this latter isolate is 9.29 Mb, suggesting that mixed colonies were sequenced. Providencia is in the *Proteus*-*Morganella* group, whereas *Citrobacter* is more closely related to *Escherichia* and *Shigella*; however, these two isolates are closely related on the tree (Appendix A). We therefore consider it likely that both are, in fact, *P. rettgeri* (Table 1).

### 3.2. Diversity by MLST and Capsule Typing

Kleborate assigned the 43 Kp strains to 38 distinct STs, all but 3 of which were represented by a single isolate. A phylogenetic tree showing the Kp STs is provided in Figure 1, and a tree for all 51 isolates is provided in Appendix A. The tree is also available to download and explore via the Microreact project at https://tinyurl.com/37xn4m62 (accessed on 1 February 2022). The most common ST among the Kp isolates is ST37, represented by four clonal isolates. A single locus variant of ST37 (ST37-1LV) clusters closely with this clone, and a double locus variant (ST37-2LV) clusters more distantly.

To compare the Kp lineages between different farms and cities, we mapped the origin of the isolates (city and farm) onto the tree on Figure 1. The five isolates corresponding to the ST37 cluster (including the ST37-1LV isolate) were each recovered from different farms, four of which were in Quetta and one from around Faisalabad. Thus, the repeated recovery of this lineage cannot be explained simply by local clonal expansion; instead, this clone has spread between farms and cities. Similarly, the two ST11 isolates were sampled from both Quetta and Sargodha.

Just as very similar isolates are noted from different farms and cities, so a single farm can harbour a high level of diversity. The most striking example of this is the SH farm in Faisalabad, from which 16 Kp isolates were sequenced. Apart from the two isolates of ST1315, every other isolate recovered from this farm corresponded to a distinct ST representing the breadth of the tree (indicated by the green bars in Figure 1). Thus, the diversity of the isolates from this single farm mirrors the diversity present in the whole dataset.

We also considered the diversity of the Kp isolates in terms of capsule type. The *wzi* gene encodes an outer membrane protein involved in cell surface attachment, and its high level of conservation means that it has been used for rapid capsule typing (K typing) [23]. The 43 Kp isolates were represented by 27 different *wzi* allele types, and the distribution of these types was broadly consistent with that observed with MLST (Appendix A). The most common *wzi* allele was *wzi*14 (5/43, 12%), which corresponded to the cluster of four isolates of ST37 and the ST37-1LV isolate. However, there are also some discrepancies between ST and *wzi* allele. For example, the two ST11 isolates corresponded to *wzi*150, but this allele was also found in the unrelated lineage ST4075. Such a pattern could either reflect homoplasy (the independent emergence of the same *wzi* lineage in different STs) or, more likely, the horizontal transfer of the *wzi* locus.

Similar patterns were noted with the serotype assignment, with the five isolates of the ST37 cluster being assigned as KL14. Serotypes K1 and K2, associated with hypervirulent strains, were not found, and there were five novel KL types (Appendix A). The most common O-type in the Kp isolates was O2, which was found in the two ST11 isolates and nine other isolates of diverse Kp lineages.

### 3.3. The Presence and Distribution of Antimicrobial Resistance Genes (ARGs)

We used Kleborate and Resfinder to detect ARGs and MOB-suite to assign these ARGs to specific plasmids (see below, and Methods). Overall, the level of resistance was low; we did not detect any carbapenemase genes or genes predicted to confer resistance to colistin. Moreover, we did not detect any ARGs corresponding to the clinically relevant antibiotic classes (AGly, Flq, Sul, Tmt, Bla,, Tet, Rif, Phe, MLS) in 35/43 (81%) of the Kp isolates or in the six isolates of other *Klebsiella* species, (Appendix A).

Two Kp isolates (ST37-1LV and ST2118) have acquired single *bla* genes (*bla*_ACT-16_ and *bla*_CphA2_, respectively) and the ST37-1LV isolate also contains the *fosA2* gene (predicted to confer fosfomycin resistance). However, more notably, five Kp isolates, representing four lineages, and the two presumptive *P. rettgeri* isolates contain multiple ARGs (Figure 2 and Appendix A). Firstly, the pair of Kp ST11 isolates harbour seven ARGs conferring resistance to aminoglycosides (*aph(3″)-Ib*, *aph(6)-Id*), quinolone (*qnrB1*), sulphonamides (*sul2)*, trimethoprim (*dfrA14*) and β-Lactamas (*bla*_TEM-1B_, *bla*_CTX-M-15_). As mentioned above, the two ST11 isolates were recovered from different cities and the observation that they have identical resistance profiles suggests that they correspond to a ST11 subvariant that may be widely disseminated in Pakistan.

The second pair of Kp isolates that share identical ARG profiles are those corresponding to ST995 and ST1391-1LV. These loosely cluster on the tree and were recovered from the same farm in Sargodha. Both harbour the ARG profile of 9 genes resistant to aminoglycosides (*aph*(*3*″)-*Ib*, *aph*(*6*)-*Id*), quinolones (*qnrB1*), sulphonamides (*sul2*), trimethoprim (*dfrA14*), β-Lactams (*bla*_SHV-106_, *bla*_TEM-1B_, *bla*_CTX-M-15_) and tetracyclines (*tet(A)*). The ST985 isolate from Quetta was found to harbour 16 ARGs to aminoglycosides (*aac(3)-IId*, *ant*(*3*″)-*Ia*, *aph*(*3*′)-*Ia*, *aph*(*6*)-*Id*), quinolones (*qnrS1*), sulphonamide (*sul3*, *sul2*), trimethoprim (*dfrA14*), β-lactams (*bla*_TEM-1B_, *bla*_SHV-187_, *bla*_CTX-M-55_), tetracycline (*tet(A)*), phenicols (*floR*), rifampicin (*ARR-3*) and macrolides (*lnu(F)*, *mph(A)*). Finally, the two presumptive *P. rettgeri* isolates harboured identical ARG profiles with resistance to aminoglycosides (*ant*(*3*″)-*Ia*, *ant*(*2*″)-*Ia*), sulphonamide (*sul1)*, quinolones (*qnrA1)*, chloramphenicol (*cmlA1)*, rifampicin (*ARR-3_4)* and β-lactams (*bla*_VEB-1_, *bla*_OXA-10_). These two isolates were recovered from the same farm in Quetta.

### 3.4. The Predicted Plasmid Carriage of ARGs

The observation that different Kp lineages share identical ARG profiles suggests that these resistance traits have been acquired via one or more mobile genetic elements, namely, plasmids. To explore the presence of plasmids in our data, we followed Gibbon et al. [24] and used MOB-suite to assign contigs either as chromosomal, or matching a plasmid within the *Enterobactericae* database (see Section 2). Analysing all 51 genomes, MOB-suite returned a total of 121 plasmids; these corresponded to 71 different accession numbers (because some plasmids were found in more than one isolate) and 19 replicon types. The most common replicon type was IncFIB(K). The distribution of the plasmids and replicon types is shown in Appendix A, respectively, although it is not possible to assign replicon types to specific plasmids. According to MOB-suite, of the 51 isolates, only 5 did not contain any plasmids. A total of 16 isolates contained 1 plasmid, 13 isolates contained 2 plasmids, 7 isolates contained 3 plasmids, 2 isolates contained 4 plasmids, 5 isolates contained 5 plasmids, 3 isolates contained 6 plasmids, and 1 isolate contained 7 plasmids.

Using MOB-suite, we assigned all contigs containing one or more ARG either to a plasmid or to the chromosome. Of the 71 different plasmids, 10 were predicted to contain at least 1 ARG; 8 of these were present in 1 of the 4 lineages containing multiple ARGs—ST11, ST985, ST995 and ST1391-1LV—with the other 2 carried by the presumptive *P. rettgeri* isolates (Figure 2). Seven of the AMR plasmids were predicted to contain multiple ARGs. For three of these plasmids, the set of ARGs were highly similar, with each sharing a core set: *sul2*, *aph*(*3*″)-*Ib_5*, *aph*(*6*)-*Id_1*, *bla*_TEM-1B_ and *bla*_CTX-M-15_. This indicates that these genes are linked on an element that has transferred between plasmids. These three plasmids are present in the multidrug-resistant Kp lineages ST11, ST1391-1LV and ST995, whilst the plasmid in ST985 contains a different set of ARGs.

The two isolates of ST11 harbour a plasmid that matches CP106925 (pCTXM15_DHQP1400954-like). In addition to the core set of ARGs listed above, this plasmid is also predicted to carry the ARGs *dfrA14* and *qnrB1*. This plasmid, with the same linked ARGs, has previously been found to be associated with diverse Kp lineages (ST1012, ST2167, ST48-1LV and ST495) recovered from clinical samples and wastewater in Quetta, illustrating that it is transmitting freely across the local environment (our unpublished data). Plasmid CP010390 (p6234-like) and plasmid CP008930 (pPMK1-A-like) are carried by the related multidrug-resistant isolates ST1391-1LV and ST995, respectively. These plasmids also contain the same core set of ARGs in addition to *tet*(A). The ST1391-1LV isolate additionally contains *qnrB1*. These two isolates were recovered from the same farm in Sargodha.

The third multidrug-resistant Kp lineage is ST985, which is predicted to harbour a plasmid related to KM198330 (pDGSE139-like). This plasmid harbours at least 11 ARGs, including the ESBL gene *bla*_CTX-M-55_ (but not *bla*_CTX-M-15_), ARR-3 and *lnu*(F). Finally, the two presumptive *P. rettgeri* isolates were predicted to carry plasmid CP017672 (pRB151-NDM-like), which harbours *ant*(*3*″)-*Ia*, *sul1*, *qnrA1*, *sul1*, *cmlA1*, *ARR-3*, *ant*(*2*″)-*Ia*, *blaVEB-1* and *bla*_OXA-10_ (Figure 2).

### 3.5. The Geographical Distribution of Common Plasmids

When considered altogether, the isolates containing multiple ARGs and associated plasmids are present in multiple farms in Quetta, Sargodha and Faisalabad. However, because AMR plasmids are only found in one or two isolates, it is not possible to infer plasmid-specific patterns of spread, both with respect to geographic region or host lineage. We addressed this by considering the distribution of the most common plasmids in our data. Although none of these common plasmids contained ARGs, their prevalence makes it possible to gauge to what extent specific plasmids are circulating between diverse *Klebsiella* lineages at the level of farm or city, or else are randomly distributed (Figure 3). The most common plasmids in the dataset were CP009275 (pKV1-like) (*n* = 14), CP011627 (pCAV1374-14-like) (*n* = 7), CP013713 (J1 plasmid 2-like) (*n* = 6) and CP011633 (pCAV1374-150-like) (*n* = 5). The most common plasmid, CP009275 (pKV1-like), is most similar to one first recorded in a nitrogen fixing strain of *K. variicola* (strain DX120E) isolated from sugar cane in China [25]. This plasmid was not present in either of the two *K. variicola* isolates in our data but was present in all four *K. quasipneumoniae* subsp *pseudopneumoniae* isolates (three from Quetta, and one from Faisalabad) and in ten Kp isolates from four different farms, all in Quetta. Whilst this plasmid has spread between different *Klebsiella* species and diverse Kp lineages on different farms, it shows a non-random distribution with respect to region, as 13/14 isolates carrying this plasmid are from Quetta.

The second, third and fourth most common plasmids in the data also show associations with cities or single farms. CP011627 is present in six diverse Kp STs and the pair of ST1315 isolates. Six of the seven isolates harbouring this plasmid were recovered from the SH farm in Faisalabad, with the exception being from a farm in Quetta. Plasmid CP013713 exhibits a very similar distribution to CP011627; it is present in six Kp isolates (including the pair of ST1315 isolates and three other isolates which also contain CP011627). Five of the six isolates harbouring CP011627 were recovered from the SH farm in Faisalbad, with the exception being from a farm in Quetta. Finally, plasmid CP011633 is present in five Kp isolates, two of which corresponded to ST37; the other three were from diverse lineages. In this case, all five isolates were recovered from three farms in Quetta. The distribution of these plasmids is summarised in Figure 3.

### 3.6. Virulence

Finally, Kleborate also identifies key virulence genes in the genomes. The only virulence gene identified was *ybt*, which encodes the siderophore yersiniabactin. Five Kp isolates harboured chromosomal copies of this gene, including ST11 (both isolates) and the ST985 isolates. These lineages harboured *ybt*14 and 16, respectively. It is of concern that these lineages harbour both virulence and resistance traits. The other two isolates to harbour *ybt* are those corresponding to ST45 and ST2876. No ARGs were identified in these isolates, which were recovered from the SH farm in Faisalabad.

In sum, the WGS data revealed high levels of diversity, evidence of local plasmid spread, and the presence of Kp lineages containing multiple ARGs.

## 4. Discussion

Kp is a priority ESKAPE pathogen, and the prevalence of Kp strains exhibiting resistance to carbapenems and other β-lactam antibiotics has increased steadily on a global scale in recent years [26]. Moreover, the ecological breadth of this species makes it a pertinent model to assess the role of nonclinical settings in the emergence and spread of AMR (the so-called ‘One-Health’ approach). This framework may be highly informative in Pakistan, where a high level of agricultural activity is combined with poor antibiotic stewardship, and Kp strains resistant to carbapenems and third-generation cephalosporins are known to be circulating in veterinary and farm settings [15,27]. However, WGS has not been deployed on large population samples of Kp from agricultural settings in Pakistan, which makes it impossible to generate a more complete picture of the diversity, the presence of plasmids and other MGES and the increased phylogenetic resolution for inferring patterns of spread [14]. Here, we begin to address this by presenting an analysis of genome data for 49 *Klebsiella* isolates (43 Kp) and 2 isolates of *P. rettgeri* from 10 dairy farms in the regions of Faisalabad, Sargodha and Quetta. The comparison of three distinct regions is important to understand whether Kp clones are expanding locally on single farms or regions, or whether there are geographical differences regarding the presence of distinct strains.

The data reveal a remarkably high level of diversity within the Kp population; 38 STs were recovered, but only three of these were represented by more than one isolate. This contrasts to comparable studies on European farms, which show much lower levels of Kp diversity due to clonal expansion on single farms [14,28,29]. This fundamental difference suggests that Kp in dairy farms in Pakistan may be more often acquired from diverse environmental sources and/or transmitted more freely between different farms or regions. It is particularly striking that the 16 Kp isolates recovered from the SH farm in Faisalabad correspond to 15 different STs. Whilst this indicates that the tracking of specific clones may not be a valid approach in agricultural settings in Pakistan, our data suggest that specific plasmids are circulating locally. An example is the plasmid CP106925 (pCTXM15_DHQP1400954-like), which carries multiple ARGs and is present in the ST11 isolates. This plasmid has also been noted to be associated with other STs recovered from clinical and environmental settings in Quetta (our unpublished data), which suggests it is circulating in multiple lineages in this region. Given such examples, it is possible that as additional data sets are generated and long-read sequencing becomes more routine, epidemiological surveillance may be more appropriately focused on specific AMR plasmids, rather than the underlying strains.

The majority (84%) of the isolates do not contain major ARGs and, importantly, we did not identify any carbapenemases or predicted colistin resistance. It is important to note that we did not select for resistance to carbapenems or any other antibiotics other than amoxicillin. This was in order to capture the diversity of the underlying Kp population and to gauge the prevalence of AMR. The low prevalence of AMR in our data should, therefore, not be interpreted as indicating that resistance is of negligible importance in these settings. Indeed, we recovered 4 Kp lineages (ST11, ST1391-1LV, ST995 and ST985) that contain between 7 and 16 ARGs, including ESBL-encoding genes. The ST11 and ST985 isolates also contain the important virulence factor *ybt*, which encodes yersiniabactin, although we did not detect the presence of any other key virulence gene, such as *iuc* or *rmp*A.

ST11 is a globally disseminated clone associated with carbapenemases and other resistance genes as well as virulence factors, particularly in Asia [30]. The presence of two isolates of ST11 that possess both multiple resistance and virulence genes is noteworthy given that this lineage has previously been reported from nonclinical settings in Pakistan [27]. We note that the two ST11 isolates in our data correspond to an unknown K-type, with the closest match being KL107. KL107 is common in the closely related healthcare-associated AMR clone ST258 [31], but it is very rare in ST11 [32], indicating that a novel subtype of this global lineage may be circulating in Pakistan. Kp ST37 is the most common clone in our data. Although we did not find this clone to be associated with multiple antibiotic resistance, it is known to be common in clinical and environmental settings globally, and is often associated with ARGs [33].

Although we targeted *Klebsiella* species in this study, we also sequenced two isolates which we consider are likely *P. rettgeri*. Both isolates are predicted to contain a plasmid originally isolated from *P. rettgeri*. *P. rettgeri* is related to insect pathogens, but is an opportunistic pathogen of humans known to cause urinary tract infections, particularly in immunocompromised patients, and is commonly resistant to multiple antibiotics [34]. The sequencing of these two isolates was serendipitous, as they share the same plasmid containing multiple ARGs, including *bla*_OXA-10_, *bla*_CTX-M-55_, *lnu*(F)—which encodes resistance to lincosamides (widely used in veterinary medicine)—and ARR-3, encoding resistance to rifampicin. In addition to being a potential public threat to humans, this species could be acting as an important reservoir of ARGs and plasmids and warrants further surveillance.

In conclusion, our study reveals a very high level of diversity among Kp strains circulating within dairy farms in Pakistan and provides evidence that this setting is a potentially significant reservoir of plasmid-borne ARGs. We acknowledge several limitations with our study. Most importantly, we were limited to short-read data and thus not able to generate closed assemblies. This has important consequences for characterising the plasmid content of the isolates. Although MOB-suite was able to identify the best matched plasmids to our contigs, without closed plasmid assemblies it is not possible to validate these inferences, directly compare the plasmids with those previously published or confirm the presence of the ARGs.

Despite these limitations, our analysis highlights the utility of whole-genome sequencing for surveying the prevalence and distribution of ARGs in dairy farms in Pakistan. The study provides valuable benchmark data, paving the way for more large-scale WGS One-Health studies in Pakistan, and raises the question as to whether surveillance is best carried out at the level of strain, ARG or AMR plasmid. The exception of ST11 aside, the high degree of Kp strain diversity uncovered in this study points to the possibility that emerging AMR carriage in dairy farms is not primarily the result of simple clonal spread, and that further focus, specifically on AMR plasmid surveillance in Pakistan, will prove informative.

## Figures and Tables

**Figure 1 antibiotics-12-00539-f001:**
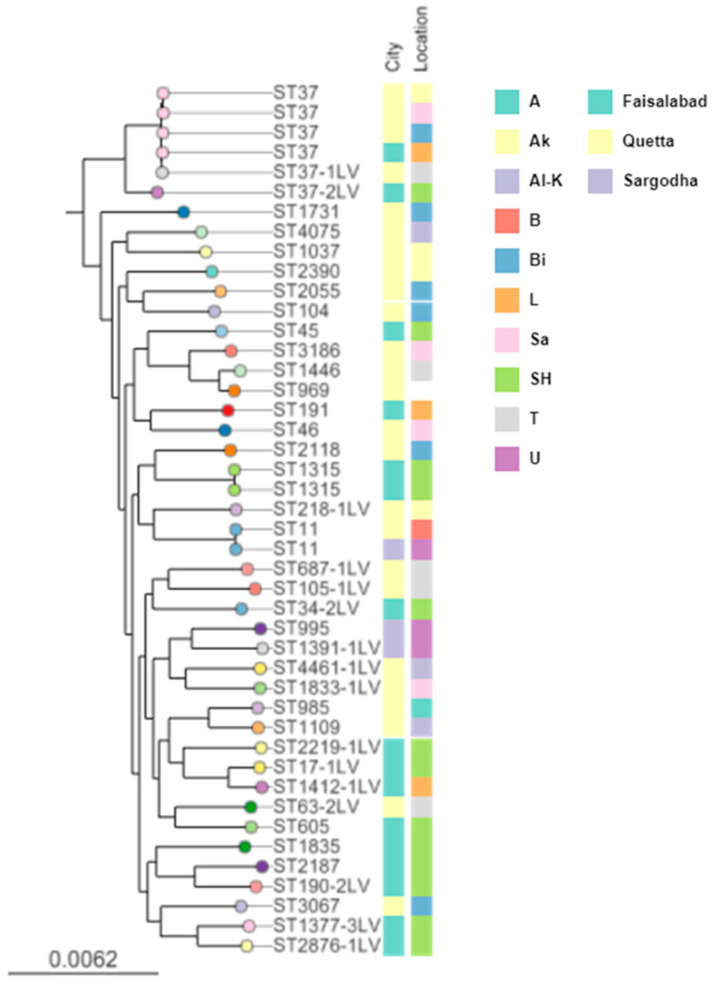
Sequence type distribution of *K. pneumoniae* among different dairy cattle farms. The location refers to the specific dairy farm. The data is available to explore via the Microreact project at https://microreact.org/project/iqM31fUDH173HKRxjpEVMi-whole-data-tree. For the exact format used in this figure, see https://microreact.org/project/7cjj9qFE8LbppJsRGXtk8f-st-distributionkp.

**Figure 2 antibiotics-12-00539-f002:**
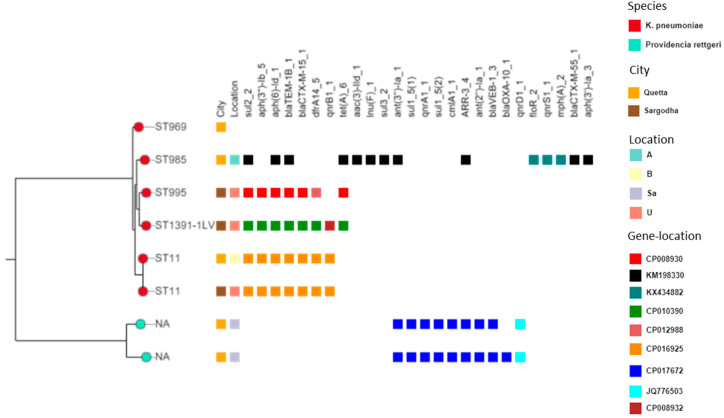
Approximate maximum-likelihood phylogenetic tree of the isolates carrying plasmids with linked resistance genes isolated from different dairy farms. The accession number of plasmids identified by MOB-suite (gene-location) and resistance genes identified using Abricate with the ResFinder database are shown. Only plasmids associated with linked resistance genes are included. See also: https://microreact.org/project/qNGWgBKZBEdnzFD6tHLuy5-linked-genes.

**Figure 3 antibiotics-12-00539-f003:**
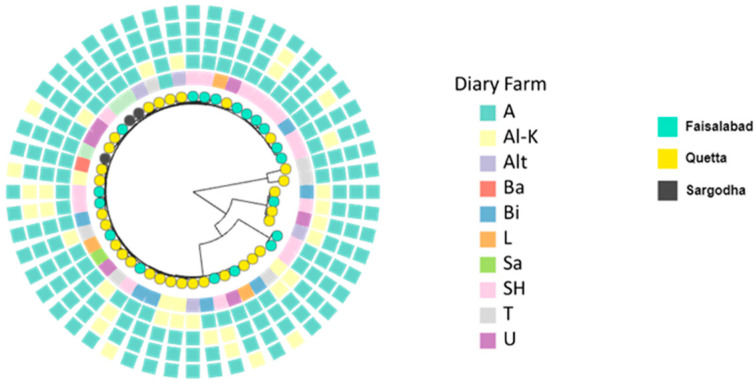
Distribution of the four most common plasmids in the *Klebsiella* isolates. The nodes indicate the region of origin (green = Faisalabad, yellow = Quetta, black = Sargodha). The inner ring indicates the specific dairy farm (legend). The next four rings indicate the presence of specific plasmids (yellow = present, green = absent). Moving outwards from the second ring, the plasmids are: CP009275, CP011627, CP013713, CP011633. The distribution of all 71 plasmids in the dataset is provided in Appendix A.

**Table 1 antibiotics-12-00539-t001:** Number of isolates corresponding to each species isolated from 10 different dairy farms in the cities of Quetta, Faisalabad and Sargodha in Pakistan.

City	Quetta							Sargodha	Faisalabad	
**Dairy Farm**	B	A	T	Al-K	Ak	Bi	Sa	U	SH	L
**Total samples**	**29**							**3**	**19**	
**Sample size location**	1	1	6	4	4	7	6	3	16	3
*K. pneumoniae*	1	1	5	3	4	6	4	3	13	3
*K. similipneumoniae*	-	-	1	1	-	1	-	-	1	-
*K. variicola*	-	-	-	-	-	-	-	-	2	-
Providencia rettgeri	-	-	-	-	-	-	2	-	-	-

## Data Availability

The genome data have been deposited under BioProject ID PRJNA935981.

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
