# Peer review of "The Diversity, Resistance Profiles and Plasmid Content of Klebsiella spp. Recovered from Dairy Farms Located around Three Cities in Pakistan"

_antibiotics, 2023, doi:10.3390/antibiotics12030539_

Round 1

Reviewer 1 Report

The diversity, resistance profiles and plasmid content of Klebsiella

spp. recovered from dairy farms located around three cities in Pakistan.” Authors: Habib et al.

Although the manuscript is mostly a collection of facts, there are some important conclusions in the paper. For example, they concluded that the repeated recovery of a lineage is due to spread between farms and cities. In addition, there is a high level of diversity within a farm. They have also tested for resistance and virulence traits.  While some strains were found to contain both traits, it is good that a majority of strains did not have major antibiotic resistance genes. Since not much sequencing data is available on bacteria from agricultural settings in Pakistan, this paper is an important addition in the field. 

I have the following minor comments on the paper: 

Please provide line numbers for easier reviewing.

Page 4 Table 1 and Figure 1: For the sake of reproducibility, it will be nice to know what the dairy farm abbreviations (B, A, T etc.) stand for.

Page 4 Line 2: “all but three of which were represented by a single isolate.”  Which isolate is this? Is that information available in Figure 1?

Figure 1: The Figure is cut off a little on the top

Figures 1 and 2: There is a “page break” sign and several paragraph signs that are showing.  They need to be removed.

Figure 1: Why is location “Ref” shown in the Figure 1 but is not there in Table 1? 

Page 5, line 38 and Page 6 Line 5: “Lactamas” I think, the authors mean, “Lactams”

Page 6, Line 19: “Analysing all 51 genomes, MOB-suite returned a total of 71 plasmids”.  For the benefit of the less-informed readers please state what is being sequenced: whole cells or isolated genomic + plasmid DNA or isolated and separated genomic and plasmid DNA.

Page 6 Line 23: “The modal average of plasmid carriage was one per isolate, but nine isolates were predicted to carry at least five plasmids.”  Since these are all definite sequence results, the word “predicted” is not understood.  The prediction is based on which part of the sequence? Also, the numbers are misleading. If nine isolates had at least five plasmids, that accounts for at least 45 plasmids. Plus there are probably many with less than five but more than one. So, probably more than half of the isolates are without plasmids. So, calculation of an average is misleading. 

Reviewer 2 Report

In the manuscript titled “The diversity, resistance profiles and plasmid content of Klebsiella spp. recovered from dairy farms located around three cities in Pakistan  authors employed the WGS approach to unravel the diversity, genetic relatedness, plasmid profiling, and antibiotic resistance pattern among the Klebsiella isolates from dairy farms in Pakistan.

Overall, the study carries a good piece of information and also has some significant data to contribute to the area of antimicrobial resistance.

Aside from that, there are a few points I'd like to bring up-

1) Please add the aim of this study in the introduction.

2) Methods, Sampling- Please add the time span of the sampling process.  

3) Methods, Sampling- What were the criteria for the selection of the animals for sample collection? What kind of cattle were selected? How dairy farms were shortlisted? Please provide information.

4) Methods, Genome Characterization- Any specific reason to use PAK-014 Kp isolate for mapping purposes. Plenty of sequences even those isolated from cattle are present in databases.

5) Methods, Genome Characterization- Is the accession number not granted yet?

6) Please correct the caption of Table S1. It is saying data of 53.

7) Table-1) What exactly are the abbreviations used in the row "dairy farm"? Please provide the full form in the caption.

8) Figure 1, 2, 3) Please remove the noise from the figures.

9) Consider adding a conclusion describing the importance, novelty, contribution, and future prospective of this work.

Reviewer 3 Report

  1. The paper titled “The diversity, resistance profiles and plasmid content of Klebsiella spp. recovered from dairy farms located around three cities in Pakistan. described whole genome sequencing to characterise 49 isolates of Klebisella spp. (including 43 Kp), and two presumptive Providencia rettgeri isolates, recovered from dairy farms located near three cities in Pakistan. The manuscrippit has potential but need following changes before consideration
  2. Title is fine and describing the manuscript in efficient manner
  3. Abstract is written good as discussed background, study objective, results as well as outcomes. Also add the different stakeholders for which the findings are useful at the end of abstract with particular focus of findings application  
  4. Introduction is written good however, add one paragraph regrading antimicrobial resistance prevelance, policy and regulations framework and its impact on human health.
  5. Introduction sentence “The WGS data revealed high levels of diversity, evidence of local plasmid spread, and the presence of Kp lineages containing multiple antimicrobial resistance genes (ARGs)” needs to be placed in the result section of the manuscripit
  6. Material and method section is written is good however, elaborate “genomes of 55 presumptive Kp from dairy cows were sequenced” with reference and detail
  7. What is the criteria for selecting cities for sampling of dairy farms?
  8. Statistical analysis is missing. Justify or add ?
  9. Results are written in a good way however add the latest references to support findings of the paper
  10. In conclusion major focus should be on findings with practical application
  11. Grammatical mistakes observed on few places so there is need to go through the paper for language and grammatical mistakes
